# The Symbiotic Bacteria—*Xenorhabdus nematophila* All and *Photorhabdus luminescens* H06 Strongly Affected the Phenoloxidase Activation of Nipa Palm Hispid, *Octodonta nipae* (Coleoptera: Chrysomelidae) Larvae

**DOI:** 10.3390/pathogens12040506

**Published:** 2023-03-23

**Authors:** Nafiu Bala Sanda, Youming Hou

**Affiliations:** 1State Key Laboratory of Ecological Pest Control for Fujian and Taiwan Crops, Department of Plant Protection, Fujian Agriculture and Forestry University, Fuzhou 350002, China; 2Department of Crop Protection, Faculty of Agriculture, Bayero University Kano, Gwarzo Road, Kano 3011, Nigeria

**Keywords:** symbiotic bacteria, proPhenoloxidase activation, *Octodonta nipae*, secondary metabolites, immune response, Entomopathogenic nematodes

## Abstract

Symbiotic bacteria form a mutualistic relationship with nematodes and are pathogenic to many insect pests. They kill insects using various strategies to evade or suppress their humoral and cellular immunity. Here we evaluate the toxic effects of these bacteria and their secondary metabolites on the survival and phenoloxidase (PO) activation of *Octodonta nipae* larvae using biochemical and molecular methods. The results show *P. luminescens* H06 and *X. nematophila* All treatments caused significant reductions in the number of *O. nipae* larvae in a dose-dependent manner. Secondly, the *O. nipae* immune system recognizes symbiotic bacteria at early and late stages of infection via the induction of C-type lectin. Live symbiotic bacteria significantly inhibit PO activity in *O. nipae* whereas heat-treated bacteria strongly increase PO activity. Additionally, expression levels of four *O. nipae* proPhenoloxidase genes following treatment with *P. luminescens* H06 and *X. nematophila* All were compared. We found that the expression levels of all proPhenoloxidase genes were significantly down-regulated at all-time points. Similarly, treatments of *O. nipae* larvae with metabolites benzylideneacetone and oxindole significantly down-regulated the expression of the PPO gene and inhibited PO activity. However, the addition of arachidonic acid to metabolite-treated larvae restored the expression level of the PPO gene and increased PO activity. Our results provide new insight into the roles of symbiotic bacteria in countering the insect phenoloxidase activation system.

## 1. Introduction

In recent years, attention has been drawn to nipa palm hispid, *Octodonta nipae* Maulik (Coleoptera: Chrysomelidae) studies especially on its immune response to pathogens such as nematodes [1,2,3,4,5], bacteria [6], and parasitoid venom [7,8]. Entomopathogenic nematodes (EPN) are roundworm-like organisms, which are found in the soil and are pathogenic to many insect pests. They are used as a biological control agent in home gardens and vegetable production against a variety of insect pests. The nematodes formed a mutual relationship with symbiotic bacteria and therefore kill insect pests by injecting toxic venom, which is released by bacteria. *Steinernema carpocapsae* and *Heterorhabditis bacteriophora* are the most used EPN as commercial biocontrol agents and are associated with symbiotic bacteria of the genera Xenorhabdus and Photorhabdus, respectively. *H. bacteriophora* and *S. carpocapsae* are the most commercially produced species for use as bio-pesticides and for research purposes [9]. The nematodes enter the insect’s hemocoel through openings such as the mouth and anus and release their symbiotic bacteria. The released symbiotic bacteria multiply and kill the host within a day or two through septicemia by secreting toxins [10]. The insect host cadaver and bacterial colonies serve as a source of food for nematodes for growth and reproduction. On the other hand, the bacteria benefit from being protected from the harmful effects of the external environment and being transmitted from one host to another [11].

Many researchers have reported that the actual killing of the insect host is mainly performed by symbiotic bacteria [12,13]. However, the nematode itself also caused mortality when isolated and infected in the target host [14,15,16]. In order to successfully kill an insect, symbiotic bacteria employ different strategies, which include the secretion of toxic substances to evade and or suppress the immune system of the target host. Herbert and Goodrich-Blair [17] reported that *X. nematophila* suppresses the host’s immune system by secreting insect toxins, cytotoxins, and hemolysins. Similarly, Ffrench-Constant et al. [18] reported that *P. luminescens* secretes pathogenicity factors such as toxic complexes (Tcs), Pir AB, Mcf1 and Mcf2, virulence cassettes (PVCs), and lytic enzymes (proteases, lipases, and phospholipases), which leads to the death of target insects within 24–48 h after injection. Prior to insect death, *P. luminescens* evades the cellular immune system by destroying the hemocytes responsible for phagocytosis. The *tca* and *tcd* complexes of *P. luminescens* were shown to have high oral toxicity to *Manduca sexta, Leptinotarsa decemlineata*, and *Bemisia tabaci* hosts [19,20]. In the same vein, XeGroEL, an insecticidal protein from *X. ehlersii*, was reported to cause high mortality of *G. mellonella* larva at 48 h post-injection [21]. Contrarily, a non-virulent strain of *Xenorhabdus* against lepidopteran insects was discovered. *X. bovienii* CS03 was found to be non-virulent to *G. mellonella* or *S. littoralis* larvae as compared to virulent strains [22].

In addition to the inhibition of cellular immune responses, symbiotic bacteria counteract the insect humoral immune system. The bacterial metabolites secreted by *P. luminescens* and *X. nematophila* were reported to inhibit the activation of the insect phenoloxidase system responsible for melanin production [18,23]. An isolated compound, Phthalic acid (PA), from *P. temperata* M1021 was found to inhibit phenoloxidase (PO) activity by 82% in *Galleria mellonella* larvae [24]. Similarly, in *P. luminescens*, (E)-1,3-dihydroxy-2-(isopropyl)-5-(2-phenylethenyl) benzene (ST), an antibiotic substance, was discovered to have inhibitory action against the *M. sexta* phenoloxidase system [25]. A bacterial metabolite, benzylideneacetone (BZA), from *X. nematophila* inhibited PO activity when injected into the larvae of *Plutella xylostella* [23]. Kim et al. [23] found that *X. nematophila* reduces PO activation by 30% compared to control reactions. Conversely, *X. innexi* fails to suppress the insect phenoloxidase activation in *G. mellonella*, *M. sexta*, and *Drosophila melanogaster* as reported by the same author [23]. The eicosanoid biosynthesis pathway is another important pathway tampered with by symbiotic bacteria [26]. This is achieved by bacterial metabolites, which have inhibitory effects against insect immune-associated phospholipase A_2_ (PLA_2_). PLA_2_ provides the initial step in eicosanoids biosynthesis responsible for non-self-signaling in insect immunity [27]. The inhibition of PLA_2_ by *X. nematophila* (benzylideneacetone) induces immune-depression (inhibition of PO activation and AMP synthesis) in *S. exigua* and *M. sexta* [23,28,29].

The pathogenicity of *P. luminescens* and *X. nematophila* was evaluated against larvae of *O. nipae*. In addition, the symbiotic effect of these bacteria on phenoloxidase activity, the expression levels of recognition, and proPO activation genes in infected larvae of *O. nipae* were tested. We further tested the toxic effects of the secondary metabolites Benzylideneacetone (BZA) and Oxindole (OXD) on PO activity and the prophenoloxidase (PPO) gene expression level in *O. nipae* larvae.

## 2. Materials and Methods

### 2.1. Insects and Chemicals

The beetles were reared as previously described by Hou et al. [30,31,32,33]. Briefly, *O. nipae* individuals of all stages were collected and reared in our laboratory before use. They were fed leaves of the fortune windmill palm, *T. fortunei* (Hook), and kept at 25 ± 1 °C, 80 ± 5% RH with a photoperiod of 12 h:12 h (light: dark) [33,34]. We purchased the eicosanoid inhibitors benzylideneacetone (BZA; trans-4-phenyl-3-buten-2-one) and oxindole (OXD; C8H9NO), the eicosanoid precursor arachidonic acid (AA:5,8,11,14-eicosatetraenoic acid), and dimethyl sulfoxide (DMSO) from the Sigma Aldrich (Shanghai, China) Trading Co., Ltd. (Sigma Aldrich Research Biochemicals, St. Louis, MO, USA).

### 2.2. Bacterial Culture and Identifications

The symbiotic bacteria *P. luminescens* H06 and *X. nematophila* All were isolated from the hemolymph of *Galleria mellonella* infected with IJs of *H. bacteriophora* H06 and *S. carpocapsae* All, respectively. Dead *G. mellonella* larvae (2–3 days after inoculation) were surface-sterilized in 70% alcohol for 10 min, flamed, and allowed to dry in a laminar airflow cabinet for 2 min. Larvae were dissected using sterile needles and scissors, and a drop of the oozing hemolymph was streaked with a needle onto nutrient agar (NBTA) plates [35]. The sealed agar plates were incubated at 28 °C for 48 h in the dark. Preliminary identification of these bacteria was performed by observing their colony morphologies. A single colony from each isolate was sub-cultured on the same medium (NBTA) to obtain uniform colonies for further identification. Subsequently, a single colony was selected and cultured on Luria–Bertani broth (LB) with shaking (600 rpm) at 28 °C for 20 h. The bacterial concentrations were determined by the adjustment of the bacterial suspension at OD_600nm_ to 0.2 using a spectrophotometer for *P. luminescens* H06 and *X. nematophila* All.

### 2.3. Survival of O. nipae Larvae Infected with P. luminescens H06 and X. nematophila All

The bioassay was conducted to determine the survival of the third instar larvae of *O. nipae* after injections with fresh cultures of symbiotic bacteria. First, 112 nl of *X. nematophila* All was injected at 2.3 × 10^6^, 2.3 × 10^5^, 7 × 10^4^, 2 × 10^4^ CFU/mL. Similar amounts of *P. luminescens* H06 were also injected into other sets of larvae at 2.9 × 10^6^, 2.9 × 10^5^, 2.9 × 10^4^, 1.2 × 10^4^ CFU/mL. The bacteria-infected larvae were placed in Petri dishes (Costar^®^, Corning Incorporated Corning, Corning, NY 14831, USA) and provided with a small piece of fortunes windmill palm, *T. fortunei*. The survival rates were checked at 6 h intervals from 12 to 60 h after treatment. For the control treatment, 112 nl of distilled water was injected into the larvae. Prior to injection, the surface of the *O. nipae* larvae was sterilized with 70% ethanol. Thirty individual larvae for each treatment were used and replicated three times to confirm the results.

### 2.4. Phenoloxidase (PO) Enzyme Activity Assay

To determine the effects of the presence of live or heat-killed *P. luminescens* H06 and *X. nematophila* All on the activity of *O. nipae* prophenoloxidase system, the hemolymph was obtained from *O. nipae* larvae as described in our previous study [5]. The hemolymph was diluted to 5:50 µL (*v*:*v*) with 50 mM phosphate buffer (pH 8.6) and centrifuged at 1700 g at 4 °C, for 1 min to obtain the supernatant. Then 0.004 g of L-Dopa (8 mmol/L L-Dopa in 10 mmol/L 50 mM phosphate buffer pH 8.6) was dissolved in 10 mL of PBS (150 mM NaCl, 2.7 mM KCl, 1.8 mM KH_2_PO_4_ and 10.1 mM Na_2_ HPO_4_; pH 8.6). Finally, 20 µL bacterial suspensions +30 µL supernatant +100 µL L-Dopa was used for the assay. For the control treatment, 20 µL of distilled water was used in place of the bacterial suspension. The relative activity of phenoloxidase was measured with spectrophotometer (SpectraMax, Molecular Devices Corporation, San Jose, CA, USA) at A 490 nm 5 min^−1^ at 20 °C.

### 2.5. Recognitions of P. luminescens H06 and X. nematophila All by O. nipae’s Immune System

To determine the ability of the *O. nipae* immune system to recognize the presence of *P. luminescens* and *X. nematophila* All in its hemocoel, we injected larvae with live bacteria from both species as described above. Samples were taken at 8, 16, and 24 h after injection. Total RNA extraction, cDNA synthesis, and qRT-PCR analysis were performed in triplicate for each biological replicate as clearly described in Sanda et al. [5]. All calculations were performed using the accompanying ABI 7500 system software with ribosomal protein S3 (rpS3) as a reference gene. The primer sequences are provided in Appendix A.

### 2.6. Effects of P. luminescens H06 and X. nematophila All Infections on O. nipae’s Prophenoloxidase Activation System

Here the aim is to determine the effects of *P. luminescens* H06 and *X. nematophila* All injections on the mRNA expressions levels of four selected genes involved in the proPO activation system of *O. nipae* larvae. These genes include Serine Protease P56 (SPP56), prophenoloxidase activation factor 1 (PPAF1), prophenoloxidase (PPO), and serine protease inhibitor 28 (SPI28) genes. Third instar larvae of *O. nipae* were injected with *P. luminescens* H06 and *X. nematophila* All as described above. Total RNA was extracted from each of the five larvae at 8, 16, and 24 h post-treatment using the TRIzol reagent (Invitrogen, Carls-bad CA, Waltham, MA, USA) as described in the manufacturer’s protocols. cDNA Synthesis and qRT-PCR analyses were performed as reported by Sanda et al. [5]. All calculations were performed using the accompanying ABI 7500 system software with ribosomal protein S3 (rpS3) as a reference gene [7]. The primer sequences are provided in the Appendix A.

### 2.7. Influence of Eicosanoid Inhibitors and Precursor on Phenoloxidase Activity in O. nipae

Benzylideneacetone (BZA) and oxindole (OXD) were dissolved in 100% dimethyl sulfoxide (DMSO) at an initial concentration of 1 M and subsequently diluted to 5mM concentrations. We determined the inhibition effects of BZA and OXD on prophenoloxidase activity in an in vitro assay using hemolymph from third instar larvae of *O. nipae* using the procedure described in our previous study [5]. Secondly, arachidonic acid (AA) was added to the treatments to determine if it can reverse the inhibition effects of these inhibitors (BZA + AA, OXD + AA). All kinetics was performed as described above. For the control treatment, only 20 µL of distilled water was added in place of various inhibitors. The relative activity of phenoloxidase was measured using a spectrophotometer (SpectraMax, Molecular Devices Corporation, San Jose, CA, USA) at A 490 nm 5 min^−1^, at 20 °C.

### 2.8. Effect of Eicosanoid Biosynthesis Inhibitor and Precursor on O. nipae PPO Gene Expression

Three treatment groups were prepared for injection into larvae at 112 nl. First, BZA and OXD only were injected individually to test their inhibitory effects on *O. nipae* PPO gene expression. Secondly, larvae were initially injected with either *P. luminescens* H06 or *X. nematophila* All. After 8 h, BZA and OXD were individually injected. Thirdly, larvae were infected with the bacteria + AA, to recover the effects of bacterial inhibition. Samples were taken at 24 h after treatments for RNA isolation. The qRT-PCR analyses were performed to ascertain the PPO gene expression level between the treatments as described above [5].

### 2.9. Statistical Analysis

All statistical analyses were performed using IBM SPSS Statistics version 22 (IBM Corporation, Armonk, NY 10504-1722, USA) (SPSS, RRID: SCR_002865). Analyses for *O. nipae* survival experiments were carried out using Kaplan–Meier tests. One-way ANOVA was used to determine the significant effect between the treatments (*p* < 0.05), and means were compared or separated using least significance differences (LSD). The level of mRNA expression of select genes in the proPO activation system of *O. nipae* at each time point was transformed by Logarithmic function and analyzed using Student’s *t*-test. Differences between mean values were analyzed and considered significant when *p* < 0.05 or considered extremely significant when *p* < 0.0001 concerning the control values.

## 3. Results

### 3.1. Survival of O. nipae Larvae Infected with P. luminescens H06 and X. nematophila All

Different bacterial doses of *P. luminescens* H06 *and X. nematophila* All were injected into the hemocoel of *O. nipae* third instar larvae. Both *P. luminescens* and *X. nematophila* All were pathogenic to the *O. nipae* larvae at all concentrations and time points post-treatment. Significant differences were observed among the *X. nematophila* (*χ*^2^ = 11.88, df = 4, *p* = 0.002, Log-Rank Test) and *P. luminescens* H06 (*χ*^2^ = 15.22, df = 4, *p* = 0.001, Log-Rank Test) concentrations for *O. nipae* larvae survival as compared to their controls. Furthermore, the results showed dose-dependent responses, with the highest *O. nipae* survival observed at lower concentrations of *X. nematophila* All (Figure 1A) and *P. luminescens* H06 (Figure 1B).

### 3.2. Inhibitory Effects of P. luminescens H06 and X. nematophila All against the Phenoloxidase Activity in O. nipae

Here, live *X. nematophila* All (*F*_2,6_ = 22.37, *p* = 0.002) and *P. luminescens* H06 (*F*_2,6_ = 19.43, *p* = 0.002) significantly inhibited the relative phenoloxidase activity in *O. nipae* humoral immune responses (Figure 2A,B). However, to further confirm the involvement of these symbiotic bacteria in the suppression of phenoloxidase activity of these larvae, we heat-treated (heat-killed) the bacteria at 100 °C for 20 min in a water bath prior to injection into the larvae. The results show a significant increase in phenoloxidase enzyme activities in heat-killed *X. nematophila* All (*F*_2,6_ = 31.14, *p* = 0.001) (Figure 2A) and *P. luminescens* H06 (*F*_2,6_ = 29.73, *p* = 0.001) (Figure 2B) treatments compared to live ones. These demonstrate that the symbiotic bacteria inhibit phenoloxidase activity to overcome the effects of the humoral immune responses of the beetle.

### 3.3. P. luminescens H06 and X. nematophila All Induce the Expression of Recognition Gene by O. nipae’s Immune System

In this study, we aim to ascertain whether the injection of *O. nipae* larvae with *P. luminescens* H06 and *X. nematophila* All will induce the expression of C-type lectin, which codes for a recognition protein in response to invasion by bacteria. We found that C-type lectin was strongly up-regulated 8, 16, and 24 h after injection of *O. nipae* larvae with *X. nematophila* All (*t*_4_ =2.91, *p* = 0.002) (Figure 3A) and *P. luminescens* H06 (*t*_4_ = 3.91, *p* = 0.001) (Figure 3B), except at 16 h after *P. luminescens* H06 treatments, which was down-regulated. Thus, the *O. nipae* larvae immune system recognizes the presence of *X. nematophila* All and *P. luminescens* H06 invasion at both early and late stages of infection via C-type lectin induction.

### 3.4. P. luminescens H06 and X. nematophila All Down-Regulate the Expression of Prophenoloxidase Genes in O. nipae

Here we investigated and compared the expression levels of four *O. nipae* proPhenoloxidase genes (including the Serine Protease P56, SPP56; prophenoloxidase activation factor 1, PPAF1; prophenoloxidase, PPO; and serine protease inhibitor 28, SPI28) between *X. nematophila* All- and *P. luminescens* H06-treated larvae at three distinct time intervals. Our qRT-PCR data reveal that SPP56 was significantly down-regulated after both *X. nematophila* All (*t*_4_ = 2.346, *p* = 0.001) and *P. luminescens* H06 challenges (*t*_4_ = 3.93, *p* = 0.03) at 8, 16, and 24 h post-injection (Figure 4A,B). Similarly, PPAF1 was down-regulated at all time points following treatments with both symbiotic bacteria except at 16 h where *P. luminescens* H06 injection resulted in significant (*t*_4_ = 3.18, *p* = 0.002) up-regulation of the gene (Figure 5A,B). Moreover, the mRNA expression levels of the PPO gene were completely down-regulated at 8, 16, and 24 h post *X. nematophila* All (*t*_4_ = 2.73, *p* = 0.03) and *P. luminescens* H06 (*t*_4_ = 2.97, *p* = 0.01) injections (Figure 6A,B). The expression level of the serine protease inhibitor SPI28, an enzyme responsible for the negative regulation of the whole protease cascade, was significantly down-regulated at all-time points upon *X. nematophila* All (*t*_4_ = 3.71, *p* = 0.02) and *P. luminescens* H06 (*t*_4_ = 3.69, *p* = 0.01) injections (Figure 7A,B). These data show that bacteria completely down-regulated the whole proPhenoloxidase enzyme activation system at all-time points.

### 3.5. Addition of Arachidonic Acid Restores the Inhibitory Effects of the Eicosanoid Inhibitors on Phenoloxidase Enzyme Activity

To test if BZA, OXD, and AA have any insecticidal activity against *O. nipae*, larvae were injected with these chemicals at 112 nl. DMSO was injected as a control treatment. Treatment with two eicosanoid biosynthesis inhibitors, BZA and OXD, showed significant insecticidal activities against *O. nipae* larvae (*F*_2,6_ = 13.91, *p* = 0.01) (Figure 8). However, no mortality of *O. nipae* larvae in AA and control treatments was recorded.

Secondly, we monitored the effects of BZA and OXD on the phenoloxidase system of the larvae in comparison with the control treatment. We found a significant decrease in the phenoloxidase enzyme activity in the hemolymph of *O. nipae* larvae treated with BZA (*F*_2,6_ = 42.13, *p* = 0.002) and OXD (*F*_2,6_ = 19.53, *p* = 0.001). However, the addition of AA rescued the defects and resulted in an increase in the phenoloxidase activity compared to the BZA and OXD treatments (Figure 9).

Similarly, we have shown above that the treatment of *O. nipae* larvae with *P. luminescens* H06 and *X. nematophila* All suppressed the expression level of four selected proPO activation genes at different time points. Here, we first injected larvae with BZA and OXD individually to ascertain their roles in the expression of the PPO gene. The results showed that the mRNA level of the PPO gene was down-regulated significantly upon treatment with BZA (*t*_4_ = 3.07, *p* = 0.002) and OXD (*t*_4_ = 4.01, *p* = 0.001) compared with the control (Figure 10). Secondly, injections of BZA into bacteria-injected larvae further suppressed the expression level of the PPO gene. There was further significant down-regulation of the PPO gene in *X. nematophila* All plus BZA (*t*_4_ = 1.817, *p* = 0.001) and *P. luminescens* H06 plus BZA (*t*_4_ = 1.31, *p* = 0.001) treatments. Surprisingly, the addition of AA to bacteria-treated larvae reverses the inhibition effects of inhibitors and bacteria plus BZA-treated larvae on PPO gene expression. The expression level of the PPO gene was highly significant in *X. nematophila* All plus AA (*t*_4_ = 1.917, *p* = 0.001) and non-significant in *P. luminescens* H06 plus AA (*t*_4_ = 2.93, *p* = 0.070) treatment.

## 4. Discussion

In the present study, we assessed the entomo-pathogenicity of *P. luminescens* H06 and *X. nematophila* All against *O. nipae* larvae hosts. Both bacteria were highly pathogenic to *O. nipae* larvae at all concentrations. However, their pathogenicity increases with an increase in the concentration, with higher concentrations causing higher mortalities of *O. nipae* larvae. The nematode–bacterium complex, however, has been successfully employed to control a wide range of invasive pests. Symbiotic bacteria were reported to be highly pathogenic to a wide range of agricultural insect pests in the laboratory [20]. According to Gerdes et al. [20], *P. luminescens* and *X. nematophila* are virulent to a large number of different insects in both laboratory and field experiments [20,36]. In *Spodoptera exigua*, different concentrations of *X. hominickii* and *X. nematophila* caused significant mortality against fifth instar larvae [37]. Similarly, Stock and Goodrich-Blair [17] reported that *X. hominickii* ANU101 was highly pathogenic to *S. exigua* and *P. xylostella*. When the two symbiotic bacteria were compared, the mortality of *G. mellonella* was significantly higher in *P. luminescens* than *X. bovienii* [36]. Similarly, up to 40 and 60% mortality of *P. xylostella* pupae were caused by *X. nematophilus* and *P. luminescens*, respectively [38]. In *Aedes aegypti*, both *P. luminescens* and *X. nematophila* caused 73% and 52% when ingested, respectively [39]. *Photorhabdus* spp. and *X. nematophila* have strong insecticidal effects on *Luciaphorus perniciosus* mites [38,40].

Insects respond to invading pathogens through pattern recognition proteins, which bind to the pathogen-associated molecular pattern (PAMP) molecules from the surface of the attaching pathogen. Several immune recognition genes are characterized in insects in response to attacks by various pathogens such as bacteria and fungi [41]. Few were specific to the immune recognition of nematodes and their symbiotic bacteria as seen from studies by Eleftherianos et al. [42], Aliota et al. [43], Kariuki, [44], and Yadav et al. [45]. Infection of *M. sexta* with *Photorhabdus* generally incites the expression of hemolin, immulectin-2, and the peptidoglycan recognition protein [42]. Similarly, the injection of insecticidal protein XeGroEL from *X. ehlersii* significantly enhanced PGRP-LB expression in *G. mellonella* [46]. Here, the expression level of C-type lectin was strongly induced by the injection of *O. nipae* larvae with *X. nematophila* All and *P. luminescens* H06. Therefore, we conclude that the immune system of *O. nipae* larvae recognizes the presence of *X. nematophila* All and *P. luminescens* H06 at both early and late stages of infection through C-type lectin induction.

Symbiotic bacterial toxins suppress the humoral and cellular immune responses such as the inhibition of phagocytosis and nodule formation, inhibition of eicosanoid biosynthesis, degradation of antimicrobial peptides, and suppression of prophenoloxidase (proPO) activation [25]. Here we determine the effects of the presence of live and heat-killed *X. nematophila* and *P. luminescens* on the activity of the *O. nipae* prophenoloxidase system. We found out that live *P. luminescens* H06 and *X. nematophila* All significantly inhibit the relative phenoloxidase enzyme activity in *O. nipae*. The heat-treated *X. nematophila* All and *P. luminescens* H06 strongly increase phenoloxidase enzyme activity compared to live bacterial treatments. This confirms the presence of certain virulence factors in the bacteria, which were destroyed after heat treatment. Similar findings were reported by several researchers including those of Yokoo et al. [47], Eleftherianos et al. [42], Song et al. [23], Rahatkhah et al. [48], and Darsouei and Karimi [49]. The decline in the PO activity was reported to be the result of certain components such as rhabduscin and lipopolysaccharides (LPS) produced by *X. nematophila* and *P. luminescens* [29]. Darsouei and Karimi [49] reported an increase in PO activity following treatment with heat-killed *X. nematophila* and *P. luminescens* in *S. exigua*.

To further confirm the involvement of some molecules produced by *P. luminescens* and *X. nematophila* in immune inhibition or suppression of host immune systems, we selected two secondary metabolites and tested their efficacies on mortality and immune responses on third instar larvae of *O. nipae* [50]. We found that BZA and OXD have significant insecticidal activities against *O. nipae* larvae. These findings are in line with the works of Jang et al. [51] and Salvadori et al. [52] that the extracellular metabolites produced by *Photorhabdus* are pathogenic to a diverse group of insects. In addition to the pathogenic effect, these eicosanoid biosynthesis inhibitors were tested on the PO activity of *O. nipae* larvae. We found significant inhibition of phenoloxidase enzyme activity in the hemolymph of *O. nipae* larvae treated with BZA and OXD compared to the control treatment. However, the addition of eicosanoid precursor AA rescued the inhibition of the phenoloxidase activity caused by BZA and OXD treatments. According to Shrestha and Kim [53], Benzylideneacetone inhibits PO activity, nodule formation, and the biosynthesis of eicosanoids by inhibiting PLA_2_ [23,28]. This is supported by Seo et al. [54] and Salvadori et al. [52], as these extracellular secondary metabolites are involved in suppressing the insect host immune system, such as inhibiting eicosanoid biosynthesis and PO activity, hemolysis, and degrading antimicrobial peptides [42]. In *Pieris rapae* larvae, benzaldehyde and its derivatives strongly inhibited PO activity [55].

In addition to suppression of PO activations by *P. luminescens* H06, *X. nematophila* All, and the two metabolites, we investigated and compared the expression levels of four *O. nipae* proPhenoloxidase genes treated with these samples. The results followed the pattern of a previous report of ours, where *S. carpocapsae* All and *H. bacteriophora* H06 suppress the activation of the *O. nipae* proPhenoloxidase system by down-regulating the expression level of four selected proPhenoloxidase enzymes as compared to the control experiment (Unpublished data). Here, the expression levels of SPP56, PPAF1, PPO, and SIP28 were significantly down-regulated in all time points after *P. luminescens* H06 and *X. nematophila* All injections. We can generally conclude that treatments with both nematodes and bacterial symbionts completely down-regulated the whole *O. nipae* proPhenoloxidase enzyme activation system at all-time points, for the successful killing of the host. To our knowledge, this is the first study that investigates the effects of these symbiotic bacteria on the expression level of proPhenoloxidase genes.

However, previous studies analyzed the suppression of humoral immune responses such as AMP synthesis and nodule formation affected by the treatments of *X. nematophila*, *P. luminescens*, and their metabolites [25,56]. According to Hwang et al. [57], the inhibition of the eicosanoid pathway by bacterial toxic factors [57] leads to the suppression of AMPs. This is confirmed by the study of Sadekuzzaman et al. [56] where the expressions of attacin-1, attacin-2, defensin, gallerimycin, and transferrin-1 of *S. exigua* were suppressed by the *X. hominickii* ANU101 treatment. Similarly, the treatment of *Photorhabdus temperata* subsp. *temperata* to *S. exigua* also inhibited the expression levels of lysozyme, gloverin, and gallerimycin [58]. In the same vein, injections of *P. temperata* subsp. *temperata* and *X. hominickii* lower the expression levels of 11 AMP genes in *S. exigua* [59]. In the present study, we treated the *O. nipae* larvae with the metabolites BZA and OXD to determine the expression of the PPO gene only. We found that the mRNA level of the PPO gene was down-regulated significantly upon treatment of both BZA and OXD compared to the control. Similarly, injections of BZA into bacteria-injected larvae further suppressed the expression level of the PPO gene. However, significant recovery of the expression level of the PPO gene was recorded upon the addition of AA to already-bacteria-treated larvae. Similarly, according to Ullah et al. [60], benzaldehyde inhibited PO activity in a concentration-dependent manner. Wang et al. [55] and Shrestha, et al. [61] reported that this compound and its derivatives inhibit PO activity and PLA_2_ synthesis, respectively. It was hypothesized by Song et al. [23] that BZA might act as a competitive inhibitor to PO because its phenylpropane core structure resembles a PO substrate, L-3,4-dihydroxyphenylalanine (L-DOPA), which suggests the inhibition of PLA_2_ activity, which, in turn, inhibits PO activation. Their results suggested that BZA treatment inhibited PO activity but showed remarkable expressions of *P. xylostella* PO gene. This is in line with our results where the BZA and OXD treatments inhibit PO activity, though in contrast with the finding that PO gene expression is inhibited by this metabolite [23].

## 5. Conclusions

In conclusion, the results of this study provide insight into a number of interesting aspects of symbiotic bacteria *O. nipae* interactions. As expected, the *P. luminescens* H06 and *X. nematophila* All treatments caused significant reductions in the number of *O. nipae* larvae. We also established the toxic effects of these symbiotic bacteria and their secondary metabolites, as well as their roles in suppressing the major cellular and humoral immune responses of *O. nipae*, such as PO activity and the expression of major proPhenoloxidase enzymes in *O. nipae* larvae, except the immune ability to recognize the presence of these bacteria. The *O. nipae* larvae immune system recognizes the presence of *P. luminescens* H06 and *X. nematophila* All invasion at both early and late stages of infection by C-type lectin induction. We conclude that our results provide new insight into the roles of *P. luminescens* H06 and *X. nematophila* All in countering insect phenoloxidase activation system.

## Figures and Tables

**Figure 1 pathogens-12-00506-f001:**
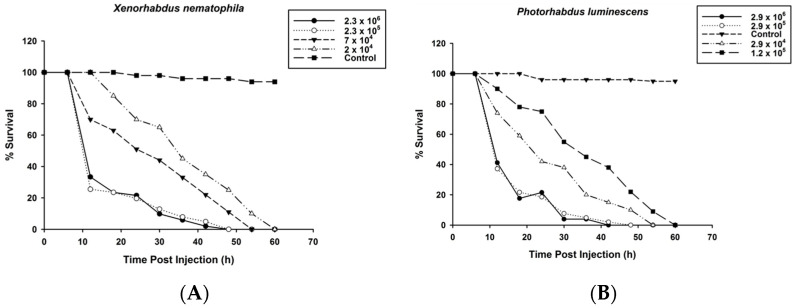
Effects of live symbiotic bacteria on survival of third instar *O. nipae* larvae at different concentrations and time points: (**A**) *X. nematophila* and (**B**) *P. luminescens*.

**Figure 2 pathogens-12-00506-f002:**
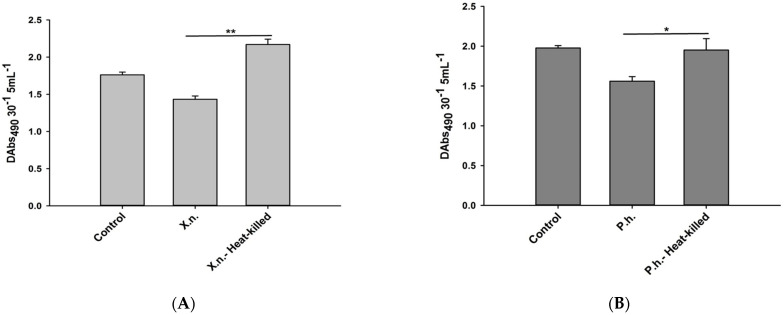
Changes in phenoloxidase activity in third instar *O. nipae* larvae treated with live and heat-killed (**A**) *X. nematophila* All and (**B**) *P. luminescens* H06. The asterisks ** (*p* < 0.001), and * (*p* < 0.01) indicate different significant levels between the control and treatments.

**Figure 3 pathogens-12-00506-f003:**
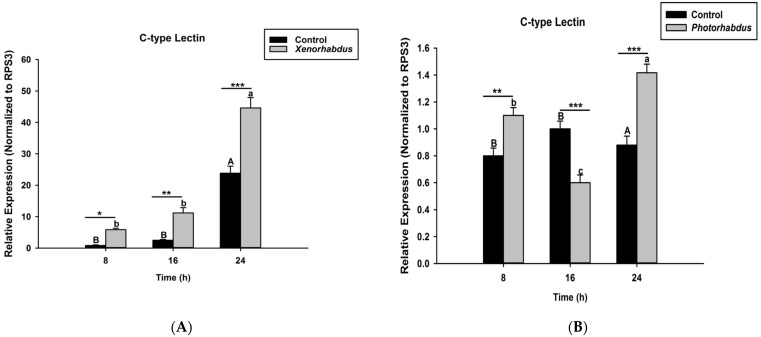
Transcription of *C-type Lectin* gene in *O. nipae* larvae infected with (**A**) *X. nematophila* All and (**B**) *P. luminescens* H06. The asterisks *** (*p* < 0.0001), ** (*p* < 0.001), and * (*p* < 0.01) indicate different significant levels between the control and bacterial treatments at the indicated time period, while “ns” indicates no significant difference.

**Figure 4 pathogens-12-00506-f004:**
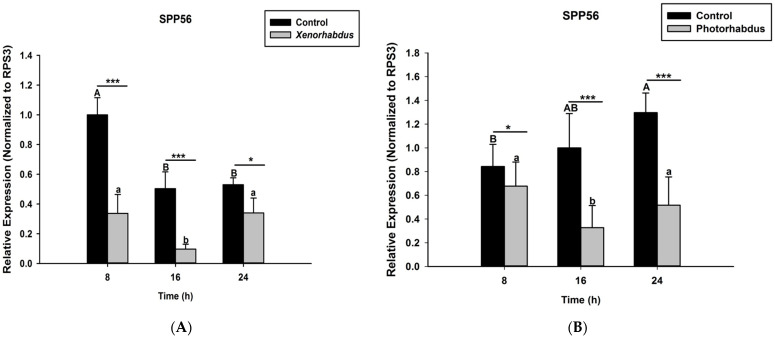
Transcription of Serine Protease P56 (SPP56) gene in *O. nipae* larvae infected with (**A**) *X. nematophila* All and (**B**) *P. luminescens* H06. The asterisks *** (*p* < 0.0001), * (*p* < 0.01) indicate different significant levels between the control and bacterial treatments at the indicated time period, while “ns” indicates no significant difference.

**Figure 5 pathogens-12-00506-f005:**
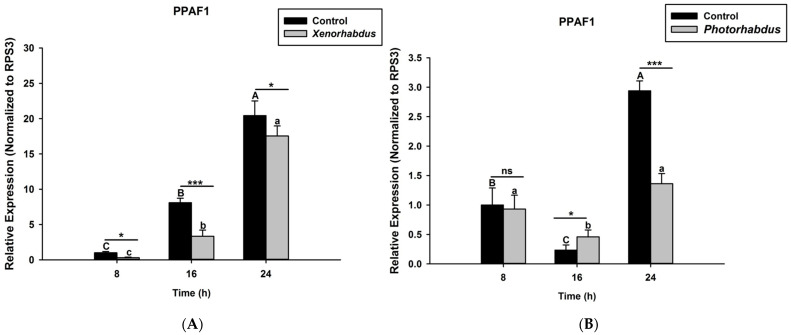
Transcription of proPhenoloxidase Activation Factor1 (PPAF1) gene in *O. nipae* larvae infected with (**A**) *X. nematophila* All and (**B**) *P. luminescens* H06. The asterisks *** (*p* < 0.0001), * (*p* < 0.01) indicate different significant levels between the control and bacterial treatments at the indicated time period, while “ns” indicates no significant difference.

**Figure 6 pathogens-12-00506-f006:**
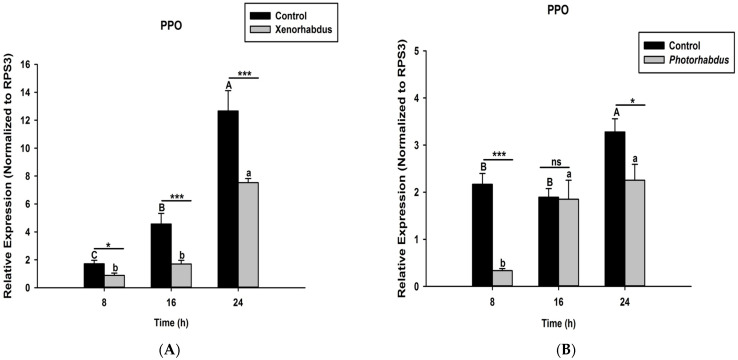
Transcription of proPhenoloxidase (PPO) gene in *O. nipae* larvae infected with (**A**) *X. nematophila* All and (**B**) *P. luminescens* H06 8, 16, and 24 h after treatments. The asterisks *** (*p* < 0.0001), * (*p* < 0.01) indicate different significant levels between the control and bacterial treatments at the indicated time period, while “ns” indicates no significant difference.

**Figure 7 pathogens-12-00506-f007:**
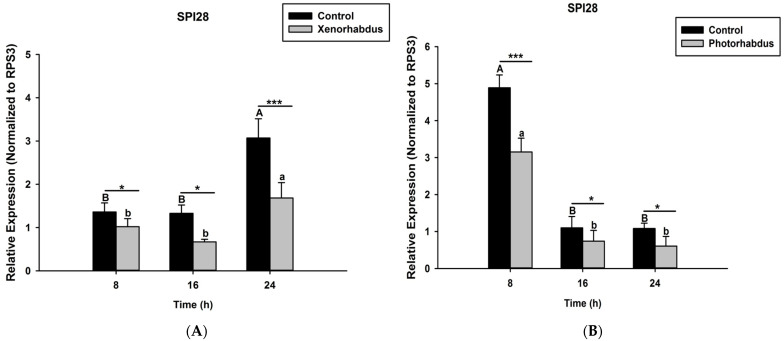
Transcription of serine protease inhibitor 28 (SPI28) gene in *O. nipae* larvae infected with (**A**) *X. nematophila* All and (**B**) *P. luminescens* H06. The asterisks *** (*p* < 0.0001), * (*p* < 0.01) indicate different significant levels between the control and bacterial treatments at the indicated time period, while “ns” indicates no significant difference.

**Figure 8 pathogens-12-00506-f008:**
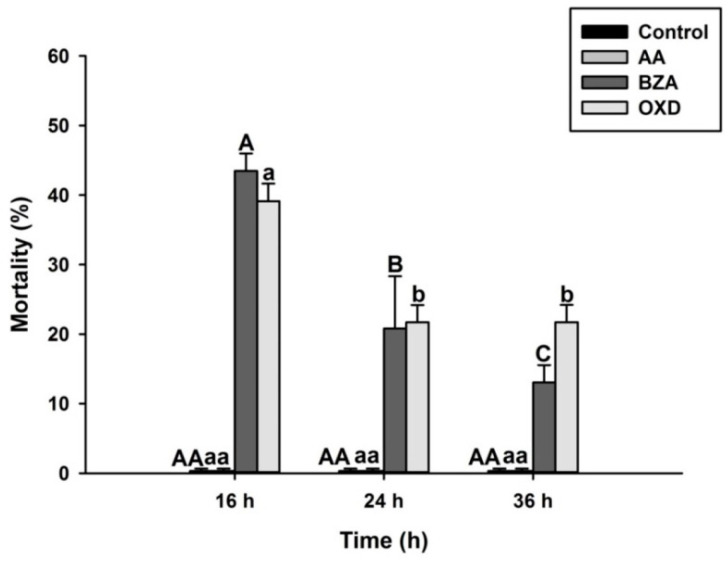
Percentage mortality of *O. nipae* larvae treated with Benzylideneacetone (BZA) and Oxindole. Different letters indicate significant differences between interactive treatments (LSD, *p* < 0.05).

**Figure 9 pathogens-12-00506-f009:**
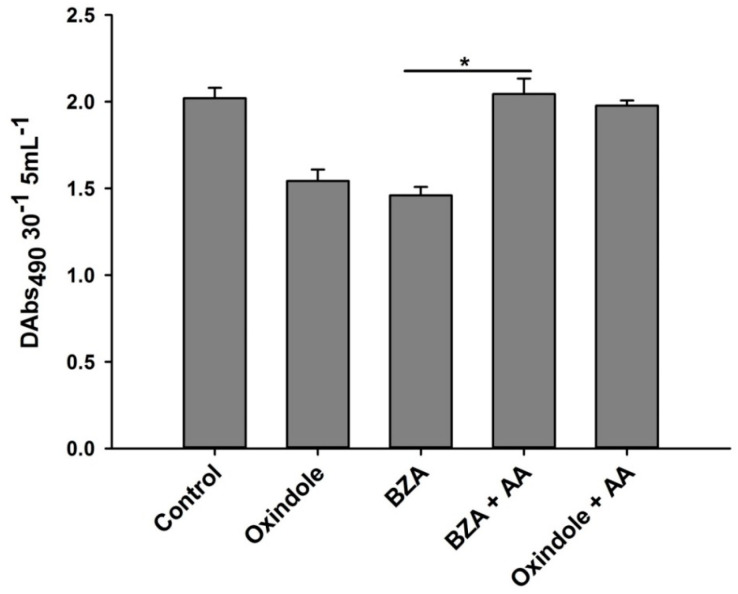
Influence of Benzylideneacetone (BZA) and Oxindole on phenoloxidase (PO) inhibition in third instar *O. nipae* larvae. The asterisks * (*p* < 0.01) indicate different significant levels between the control and treatments.

**Figure 10 pathogens-12-00506-f010:**
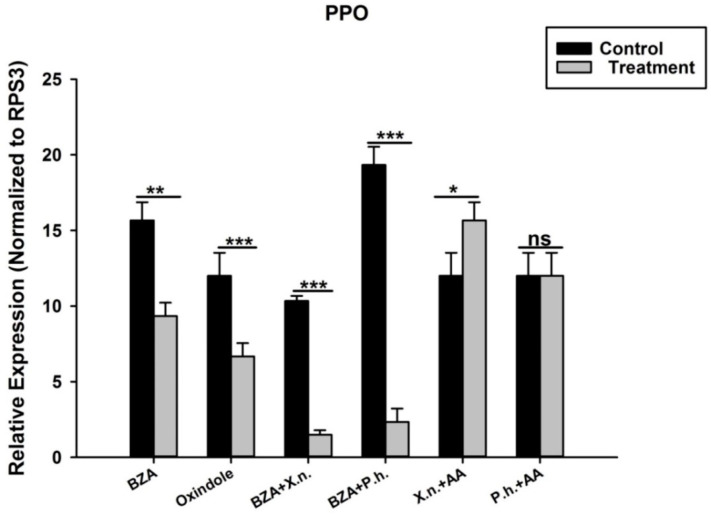
Transcription of prophenoloxidase (PPO) gene in *O. nipae* larvae injected with Metabolites. The asterisks *** (*p* < 0.0001), ** (*p* < 0.001), and * (*p* < 0.01) indicate different significant levels between the control and treatments at the indicated time period, while “ns” indicates no significant difference.

## Data Availability

Not applicable.

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
