# Peer review of "The Symbiotic Bacteria—Xenorhabdus nematophila All and Photorhabdus luminescens H06 Strongly Affected the Phenoloxidase Activation of Nipa Palm Hispid, Octodonta nipae (Coleoptera: Chrysomelidae) Larvae"

_pathogens, 2023, doi:10.3390/pathogens12040506_

Round 1
Reviewer 1 Report
The paper has great scientific merit, is very well written, and the authors are to be congratulated.

Author Response
Responses to reviewers’ questions/comments
Dear Editor,
Thank you for your letter concerning our manuscript titled “The Symbiotic bacteria; Xenorhabdus nematophila All and Photorhabdus luminescens H06 strongly affected the phenoloxidase activation of Nipa palm hispid, Octodonta nipae (Coleoptera: Chrysomelidae) larvae” (pathogens-2210384). Authors would like to thank the editorial board and reviewers who took their valuable time to provide us with constructive criticisms/comments which were found very helpful for revising and improving our manuscript. Please find below our point-by-point responses to the reviewers’ questions and comments.
Reviewer 1
Line 15
Response:
Thank you for your careful review. We have generally revised the manuscript as suggested for better understanding.
Line 37-43
Response:
The text of this paragraph was carefully reworded and restructured.
Line 77
Response:
The genus name “Galleria” replaced letter G
Line 93-97
Response:
The paragraph was restructured accordingly
Lin 124
Response:
The italic font was removed.
Line 207-209
Response: Revised accordingly
Figure 1
Response:
Thank you the good suggestion, however, the markers for each variables including the control treatment were generated automatically by the graph making software used.
Line 332-334
Response:
The bacteria was injected into larvae as mentioned in line 127 above, thank you.
Line 358-363
Response:
The paragraph was revised accordingly.
Your comments and careful review have been found very helpful in improving our manuscript. Thank you so much.
Reviewer 2 Report
Comments of mine are the same as those found in the attachment that I sent as well. Same for the Editors and the Authors, no secret. Briefly: I'll endorse and suggest from my side for publication - with minor corrections. But insist on 3 corrections. (1) INTRODUCTION: please correct the wrong and irritating interpretation of Ref [10]. The TOXINS which kill the insect, and are produced PRIOR TO killing the insects, and a few molecules of which is enough to kill the insect, are NOT ANTIBIOTICS. Antibiotics (antimicrobial peptides, non-ribosomal templated or biosynthesized NR-AMPs)) are produced only by the phase 1 EPBs and are produced throughout life by them. 2. MATERIALS AND METHODS: please give the strain names of the bacterium species name, it is especially essential for the Photorhabdus luminescens, since this species has been comprised of three physiologically, genetically, and biochemically different subspecies: P.l. ssp. luminescens, P.l. laumondii, and P.l. akhurstii. (3) IMPROVE ENGLISH, by correcting the whole paper with the help of GRAMMARLY. I made it with their otherwise excellent ABSTRACT and in the above-mentioned attachment file. I suggest accepting after the suggested minor corrections.

Author Response
Responses to reviewers’ questions/comments
Dear Editor,
Thank you for your letter concerning our manuscript titled “The Symbiotic bacteria; Xenorhabdus nematophila All and Photorhabdus luminescens H06 strongly affected the phenoloxidase activation of Nipa palm hispid, Octodonta nipae (Coleoptera: Chrysomelidae) larvae” (pathogens-2210384). Authors would like to thank the editorial board and reviewers who took their valuable time to provide us with constructive criticisms/comments which were found very helpful for revising and improving our manuscript. Please find below our point-by-point responses to the reviewers’ questions and comments.
Reviewer 2
This is a high-standard quality publication-candidate paper with invaluable useful results of the
research field, which I intend to endorse and suggest for publication. I have only three little irritating but curable things that I suggest to take into your consideration before submitting the final draft:
1. INTRODUCTION: Ref. 10 is incorrectly interpreted in the INTRODUCTION. The EPB cells
in the gut of the IJ are in the primary phase (10), therefore they produce antimicrobials
(antibiotics), AMPs) are produced NOT only „PRIOR TO” killing the insects but throughout
their life. But EPB-s - both the primary(10), and secondary-phase (20) phase ones produce
INSECT KILLING TOXINS, which are NOT ANTIBIOTICS, which are PROTEINS
(polypeptides) and a few molecules of which are enough to kill the insect. So please DO NOT
mix antibiotics with toxins, because such an irritating misleading sentence may reduce the
scientific quality of your paper providing the false impression that the authors do not familiar
with the system. But this small thing can easily to corrected.
Response:
Thank you for your careful review. We have generally revised the paragraph as suggested for better understanding.
MATERIALS AND METHODS: I miss the used Photorhabdus luninescens strain
names and Xenorhabdus nematophila species. Please add them. It is especially important in
the case of Photorhabdus luminescens because this species is comprised of 3 physiologically
and biochemically rather different subspecies, P.l. luminssecens, P.l, laumondii, and P.l.
akhurstii
Response:
The strain/subspecies names corrected as provided the author who supplied us with the nematode samples used in our experiment. The name was generally used throughout the manuscript. Some of their references include;
You J, Liang S, Cao L, Liu X, Han R. Nutritive significance of crystalline inclusion proteins of Photorhabdus luminescens in Steinernema nematodes. FEMS Microbiol Ecol. 2006 Feb;55(2):178-85. doi: 10.1111/j.1574-6941.2005.00015.x. PMID: 16420626.
Qiu, X., Yan, X., Liu, M., & Han, R. (2012). Genetic and proteomic characterization of rpoB mutations and their effect on nematicidal activity in Photorhabdus luminescens LN2.
The English should be moderately improved, for instance, by using Grammarly. In order to
help with the correction to the authors, I went through your (otherwise) excellent Abstract and
present it here for their service.
Response:
The manuscript have been checked and revised by a native English-speaking colleague.
Your comments and careful review have been found very helpful in improving our manuscript. Thank you so much.